# Circulating neprilysin hypothesis: A new opportunity for sacubitril/valsartan in patients with heart failure and preserved ejection fraction?

Josep Lupón[1,2,3], Evelyn Santiago-Vacas[1], Germán Cediel[1], Pau Codina[1], Mar Domingo[1], Elena Revuelta-López[3,4], Elisabet Zamora[1,2,3], Giosafat Spitaleri[1], Javier Santesmases[1,2], Julio Núñez[3,5,6], Antoni Bayes-Genis[1,2,3]*

1 Heart Failure Unit, Hospital Universitari Germans Trias i Pujol, Badalona, Barcelona, Spain, 2 Department of Medicine, Universitat Autònoma de Barcelona, Barcelona, Spain, 3 CIBER Cardiovascular, Instituto de Salud Carlos III, Madrid, Spain, 4 Heart Failure and Cardiac Regeneration (ICREC) Research Program, Health Science Research Institute Germans Trias i Pujol (IGTP), Badalona, Spain, 5 Cardiology Department, Hospital Clínico Universitario, INCLIVA, València, Spain, 6 Department of Medicine, Universitat de València, València, Spain

* abayesgenis@gmail.com

**Data Availability Statement:** All relevant data are within the paper and its Supporting Information files.

## Abstract

### Background

Circulating Neprilysin (sNEP) has emerged as a potential prognostic biomarker in heart failure (HF). In PARAGON-HF benefit of sacubitril/valsartan was only observed in patients with left ventricular ejection fraction (LVEF) ≤57%. We aimed to assess the prognostic value of sNEP in outpatients with HF and LVEF >57%, in comparison with patients with LVEF ≤57%.

### Methods

Consecutive HF outpatients were included from May-2006 to February-2016. The primary endpoint was the composite of all-cause death or HF hospitalization and the main secondary endpoint was the composite of cardiovascular death or HF hospitalization. For the later competing risk methods were used.

### Results

sNEP was measured in 1428 patients (age 67.7±12.7, 70.3% men, LVEF 35.8% ±14), 144 of which had a LVEF >57%. sNEP levels did not significantly differ between LVEF groups (p = 0.31). During a mean follow-up of 6±3.9 years, the primary endpoint occurred in 979 patients and the secondary composite endpoint in 714 (in 111 and 84 of the 144 patients with LVEF >57%, respectively). sNEP was significantly associated with both composite endpoints. Age- and sex- adjusted Cox regression analyses showed higher hazard ratios for sNEP in patients with LVEF >57%, both for the primary (HR 1.37 [1.16–1.61] vs. 1.04 [0.97–1.11]) and the secondary (HR 1.38 [1.21–1.55] vs. 1.11 [1.04–1.18]) composite endpoints.

**Funding:** The authors received no specific funding for this work.

**Competing interests:** A. Bayes-Genis has received speaker fees from Novartis; Julio Núñez has received speaker fees from Novartis, Vifor Pharma, Boehringer Ingelheim, Astra Zeneca, Rovi, and Novonordisk; A. Bayes-Genis and J. Lupón have applied for a patent for sNEP as prognostic biomarker and report a relationship with Critical Diagnostics. The rest of the authors have no conflicts of interest. This does not alter our adherence to PLOS ONE policies on sharing data and materials.

## Conclusions

sNEP prognostic value in patients with HF and LVEF >57% outperforms that observed in patients with lower LVEF. Precision medicine using sNEP may identify HF patients with preserved LVEF that may benefit from treatment with sacubitril/valsartan.

## Introduction

Neprilysin has become a focus of interest in cardiology [1], due to the impressive benefits of combining neprilysin inhibition and angiotensin receptor blockade demonstrated in the PARADIGM-HF trial in patients with heart failure (HF) and reduced left ventricular ejection fraction (LVEF) (HFrEF) [2]. In the cardiovascular system, neprilysin cleaves numerous vasoactive peptides. Some of these peptides have vasodilating effects (including natriuretic peptides, adrenomedullin, and bradykinin), and others have vasoconstrictor effects (angiotensin I and II, and endothelin-1, among others) [3].

Neprilysin serum levels (sNEP) exhibited significant prognostic value in heart failure (HF). At present, data on sNEP have suggested that it may play a prognostic role in both chronic [4, 5] and acutely decompensated HF [6, 7]. Moreover, sNEP might even be superior to NT-proBNP as a surrogate prognostic biomarker of the neurohormonal axis in HF [8]. However, in patients with HF and preserved LVEF (HFpEF) results were controversial [9, 10], maybe due to different sNEP quantification methods. In point of fact, not only the prognostic role of sNEP has been controversial, but also blood sNEP concentrations have also been very heterogeneous, with large differences among studies [4, 9, 11], some studies showing lower levels in HFpEF than in controls [11] and other showing higher levels in HFpEF than in HFrEF patients [4]. The correct quantification of sNEP remains a challenge that needs to be overcome to suppress potential biases regarding the interpretation of the different studies [12]. Interestingly, some sNEP quantification methods showed that circulating sNEP was catalytically active [13].

In the PARADIGM-HF study benefit of sacubitril/valsartan was observed in patients with HFrEF [2], while in the PARAGON-HF in patients with HFpEF (LVEF >45%) benefit was only observed in patients with LVEF ≤57% [14]. Indeed, in a combined analysis of PARADIGM and PARAGON, sacubitril/valsartan showed to be superior to active comparator (enalapril or valsartan) when LVEF was ~ <57% in the total cohort and ~ 62% in women [15].

Currently, HFpEF remains orphan of proven therapeutics [15]. Consequently, therapies for HFpEF are directed toward symptom management and cardiovascular risk factors. The fact that sacubitril/valsartan did not show benefit in patients with LVEF >57% in the PARAGON study does not necessarily mean that some of these patients actually did actually respond. The truth is that global results of randomized clinical trials are the balance between patients who benefit and patients who actually might have been harmed. It could be plausible from a pathobiological perspective that the response to sacubitril/Valsartan might depend on the blood concentrations of sNEP. We hypothesize that serum sNEP prognostic role might be differential across LVEF. So, in the present study we aimed to assess the prognostic value of sNEP in ambulatory patients with HFpEF and LVEF >57% (group 1), in comparison with patients with LVEF ≤57% (group 2).

## Material and methods

### Study population

From May 2006 to February 2016, ambulatory patients treated at a multidisciplinary HF clinic were consecutively included in the study. Referral inclusion criteria and blood sample

collection were described elsewhere [4]. Blood samples were obtained between 09:00 am and 12:00 pm and stored at -80˚ and analyzed without previous freeze-thaw cycles. Analyses were performed in two time periods: June-July 2014 in the first 1069 patients and November 2018 in the rest.

All participants provided written informed consent, and the local ethics committee (Comitè d'Ètica de la Investigació de l'Hospital Universitari Germans Trias i Pujol) approved the study (ethic code REGI-UNIC PI-18-037). All study procedures were in accordance with the ethical standards outlined in the Helsinki Declaration of 1975, as revised in 1983.

**Follow-up and outcomes.** All patients were followed at regular pre-defined intervals, with additional visits as required in the case of decompensation. The regular visitation schedule included a minimum of quarterly visits with nurses, biannual visits with physicians, and elective visits with geriatricians, psychiatrists, and rehabilitation physicians [4]. Patients who did not attend the regular visits were contacted by telephone.

The primary outcome was a composite of all-cause death or HF hospitalization. All-cause death, cardiovascular death, HF hospitalization and the composite of cardiovascular death or HF hospitalization were also explored as secondary outcomes. A death was considered cardiovascular in origin if it was caused by HF (decompensated HF or treatment-resistant HF in the absence of another cause), sudden death (unexpected death, witnessed or not, of a previously stable patient with no evidence of worsening HF or any other cause of death), acute myocardial infarction (directly related in time with acute myocardial infarction due to mechanic, hemodynamic, or arrhythmic complications), stroke (associated with recently appearing acute neurologic deficit), procedural (post-diagnostic or post-therapeutic cardiovascular procedure death), and other cardiovascular causes (e.g., rupture of an aneurysm, peripheral ischemia, or aortic dissection). Hospitalizations were identified from the clinic records of patients with HF, hospital wards, and the electronic Catalan history record. Twenty-one patients moved to other Spanish regions and were adequately censored for hospitalization analysis. Fatal events were identified from the clinical records of patients with HF, hospital wards, the emergency room, general practitioners, and by contacting the patient's relatives and adjudicated by an ad hoc committee (JL, M de A, BG, and MD; PM resolved possible discrepancies). Data were verified by the databases of the Catalan and Spanish Health Systems and the Spanish National Death Registry (INDEF). Follow-up was closed at October, 31, 2019.

## Neprilysin assay

Human NEP was measured using a modified sandwich immunoassay (HUMAN NEP/CD10 ELISA KIT, Aviscera Biosciences, Santa Clara, USA, Ref. SK00724-01, Lot No. 20111893). Several modifications were made to improve the analytical sensitivity of the method and obtain a lower limit of sample quantification, as reported elsewhere [4]. The modified protocol displayed analytical linearity for 0.250 to 4 ng/mL. Samples with concentrations higher than 4 ng/mL were diluted to a final concentration between 0.250 and 64 ng/mL. At a positive control value of 1.4 ng/mL, the intra- and inter-assay coefficients of variation were 3.7% and 8.9%, respectively.

## Statistical analysis

Categorical variables were expressed as percentages. Continuous variables were expressed as means (standard deviation [SD]) or medians (quartile Q1-Q3) according to normal or non-normal distributions. Normal distribution was assessed with normal Q-Q plots. Age- and sex-adjusted multivariable Cox regression analyses were performed. To fulfill the assumption of linearity the logarithmic functions of sNEP were used in the Cox models and for HR

calculation 1SD increase was used. In patients with sNEP levels below the lower range of detection (0.250 ng/mL), a concentration of 0.249 ng/mL was introduced as a continuous variable. In all analyses not involving all-cause death (secondary composite end-point, cardiovascular death and HF hospitalization), competing risk strategy by Gray method was adopted, considering non-cardiovascular death as the competing event for the secondary composite end-point and cardiovascular death and any death for HF-related hospitalization.

Statistical analyses were performed using SPSS 24 (SPSS Inc., Chicago, IL), including the R package by Bob Gray for SPSS and STATA V.13.0 (College Station, Texas, USA). A two-sided p<0.05 was considered significant.

## Results

Circulating sNEP was measured in 1,428 patients with HF who were consecutively enrolled in the study from May 2006 to February 2016, out of the 1,765 patients who were attended during this period time. No clinical criteria for exclusion were established and only consent and availability of blood sample determined the included patients. Table 1 shows the baseline characteristics of the cohort based on LVEF (≤57%, N = 1284 vs. >57%, N = 144). In summary, mean age of the total cohort was 67.7±12.7 years, 70.3% were men, in 48.9% etiology of HF was ischemic heart disease and mean LVEF was 35.8% ±14. Significant differences were found between patients with LVEF ≤57% and >57%. Patients with LVEF >57% were older, predominantly women from hypertensive or valvular etiologies and were in worse NYHA functional class. Remarkably, sNEP levels did not significantly differ between the two groups of patients (p = 0.31), while NTproBNP was significantly higher in patients with LVEF < 57% (p<0.001).

During a mean follow-up of 6 ± 3.9 years, 856 patients died; 459 deaths were from cardiovascular causes (53.6%), 344 from non-cardiovascular causes (40.2%), and 53 of unknown causes (6.2%). Among known cardiovascular causes of death, the main causes were refractory HF in 246 (53.6%) patients, sudden death in 107 (23.3%), and acute myocardial infarction in 32 (7%) patients. Additionally, 523 patients (36.6%) were admitted to the hospital for HF during the follow-up. Still, 979 patients (68.6%) fulfilled the primary endpoint of all-cause death or HF hospitalization and 714 (51.1%) the main secondary composite end-point of cardiovascular death or HF hospitalization. As shown in Table 2 all endpoints occurred more frequently in patients with LVEF >57%.

### sNEP and outcomes

As continuous variable sNEP values were significantly associated with all endpoints in patients with LVEF >57%; only with cardiovascular death and the secondary composite endpoint in patients with LVEF ≤57% (Fig 1). Age- and sex- adjusted Cox regression analyses showed higher hazard ratios (HR) for sNEP in patients with LVEF >57% for all the endpoints, being statistically significant the interaction with LVEF category for the primary endpoint of all-cause death or HF hospitalization and for all-cause death (Fig 1). Fig 2 shows the age- and sex- adjusted event-free survival curves for the primary endpoint (A) and incidence curves for the secondary composite endpoint (B) relative to LVEF group and levels of sNEP above/below the median.

Harrell's C-statistic for sNEP above/below the median, taking also in consideration age and sex, was 0.730 (0.682–0.779) in patients with LVEF >57% and 0.677 (0.664–0.776) in patients with LVEF ≤57% for the primary composite end-point, and 0.704 (0.643–0.765) and 0.629 (0.603–0.655), respectively, for the secondary composite end-point of cardiovascular death or HF hospitalization.

**Table 1. Demographic and clinical characteristics.**

| | LVEF > 57% | LVEF ≤ 57% | p-value |
|---|---|---|---|
| | N = 144 | N = 1284 | |
| Age, years | 70.1 ± 15 | 67.4 ± 12.4 | <0.001 |
| Male gender, n (%) | 63 (43.8) | 941 (73.3) | <0.001 |
| Etiology, n (%) | | | <0.001 |
| Ischemic HD | 17 (11.8) | 682 (53.1) | |
| Dilated CM | 6 (4.2) | 179 (13.9) | |
| Valvular | 37 (25.7) | 117 (9.1) | |
| Hypertensive | 40 (27.8) | 100 (7.8) | |
| Alcohol induced CM | 3 (2.1) | 66 (5.1) | |
| Others | 41 (28.5) | 140 (10.9) | |
| HF duration, months | 21.4 (4.3–58.2) | 14.4 (2–60) | 0.12 |
| Hypertension | 105 (72.9) | 817 (63.6) | 0.03 |
| Diabetes | 61 (42.4) | 518 (40.3) | 0.64 |
| COPD | 24 (16.7) | 213 (16.6) | 0.98 |
| NYHA class, n (%) | | | <0.05 |
| I | 14 (9.7) | 66 (5.1) | |
| II | 77 (53.5) | 915 (71.3) | |
| III | 51 (35.4) | 295 (23.0) | |
| IV | 2 (1.4) | 8 (0.6) | |
| BMI, kg/m$^2$ | 27.6 (24–33.2) | 26.9 (24.2–30.3) | 0.04 |
| sNEP, ng/mL | 0.601 (0.25–1.47) | 0.556 (0.25–0.99) | 0.31 |
| NT-proBNP, pg/mL | 861 (257.7–2758) | 1564 (676–3626) | <0.001 |
| LVEF, % | 65.4 ± 5.9 | 32.5 ± 10.2 | <0.001 |
| HF treatments (follow-up), n (%) | | | |
| Beta-blocker | 114 (79.2) | 1192 (92.8) | <0.001 |
| ACEI or ARB | 100 (69.4) | 1163 (90.6) | <0.001 |
| Sacubitril/valsartan | 0 | 119 (9.3) | <0.001 |
| MRA | 75 (52.1) | 876 (68.2) | <0.001 |
| Ivabradine | 4 (2.8) | 294 (22.9) | <0.001 |
| Loop diuretic | 127 (88.2) | 1193 (92.9) | 0.04 |
| Digoxin | 61 (42.4) | 543 (42.3) | 0.99 |
| Hydralazine | 76 (52.8) | 499 (38.9) | 0.001 |
| Nitrates | 68 (47.2) | 722 (56.2) | 0.04 |
| CRT | 3 (2.1) | 171 (13.3) | <0.001 |
| ICD | 8 (5.6) | 238 (18.5) | <0.001 |

ACEI, angiotensin-converting enzyme inhibitor; ARB, angiotensin receptor blocker; BMI, body mass index; CM, cardiomyopathy; COPD, chronic obstructive pulmonary disease; CRT, cardiac resynchronization therapy; ICD, implantable cardiac defibrillator; HF, heart failure; LVEF, left ventricular ejection fraction; MRA, mineralocorticoid receptor antagonist; NT-proBNP, N-terminal pro-brain natriuretic peptide; NYHA, New York Heart Association; sNEP, soluble Neprilysin serum levels.

## Discussion

Our data show a relevant prognostic value of sNEP in patients with LVEF >57% and open a new avenue of personalized treatment in HFpEF patients with really preserved LVEF. Current therapies for HFpEF are directed toward symptom management and cardiovascular risk factors due to failure of all trials conducted to date [16]. However, the global effect obtained in a clinical trial is likely the sum of beneficial effect in some patients and neutral or negative effects

**Table 2. End-points distribution based on LVEF group.**

|  | LVEF > 57% | LVEF ≤ 57% | p-value |
|---|---|---|---|
|  | N = 144 | N = 1284 |  |
| Primary end-point* | 111 (77.1) | 868 (67.6) | 0.020 |
| All-cause death | 103 (71.5) | 753 (58.6) | 0.003 |
| Cardiovascular death | 60 (41.7) | 399 (31.1) | 0.001 |
| HF hospitalization | 70 (48.6) | 453 (35.3) | 0.002 |
| Secondary composite end-point# | 84 (58.3) | 630 (49.1) | 0.068 |

*Primary end-point: all-cause death or heart failure hospitalization.

#Secondary composite end-point: cardiovascular death or heart failure hospitalization.

in other, thus resulting in a non-significant benefit. Thus, a neutral global result does not necessarily imply that no patient may benefit of the active treatment. The fact that sacubitril/valsartan did not show benefit as a whole in patients with LVEF >57% in the PARAGON trial [14] does not necessarily mean that some of these patients actually do.

sNEP demonstrated significant prognostic value in HF, both in chronic and acutely decompensated HF [4–7]. However, in patients with HF and preserved LVEF (HFpEF) results were controversial. Goliasch et al. did not find a correlation of sNEP with the combined endpoint of cardiovascular death or HF hospitalization in a registry of 144 HFpEF (LVEF ≥50%) patients [9]. We previously reported sNEP circulating levels to be associated with outcomes in a cohort of consecutive ambulatory patients with HF [4]. Most of our patients had HFrEF, but a subgroup of patients (n = 184) had LVEF >45%. In this HFpEF cohort we found that sNEP levels were prognostically meaningful with both the composite primary endpoint of cardiovascular death or HF hospitalization [4]. The disparities between the two studies could be due to clinical

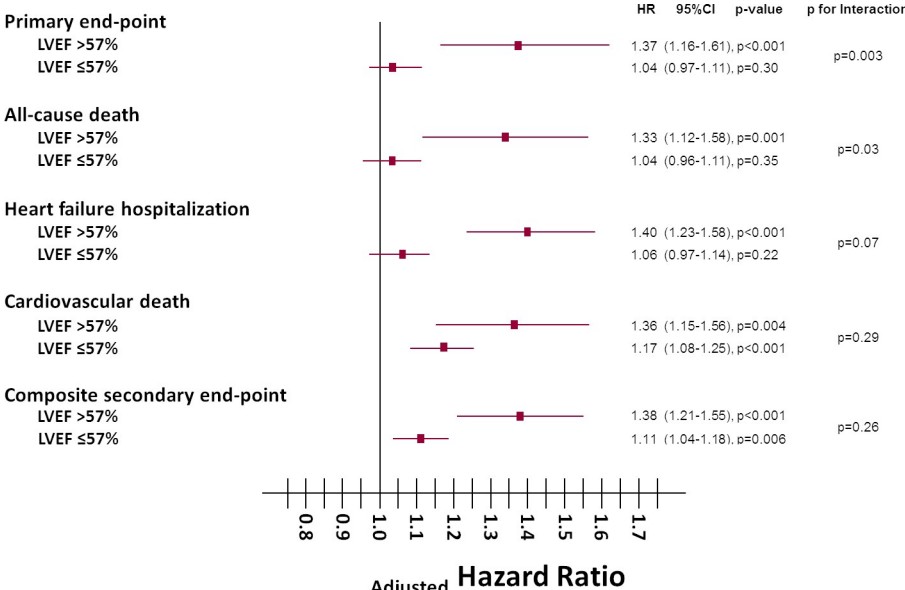

**Fig 1. Age and sex-adjusted hazard ratios for all end-points based on sNEP levels as continuous variable and on LVEF group.** Core brands represents hazard ratio and lines track spread from lower to upper 95% confidence interval. sNEP levels were log-transformed and standardized to be interpreted by 1 SD. Primary composite end-point: all-cause death or HF hospitalization. Secondary composite end-point: cardiovascular death or HF hospitalization.

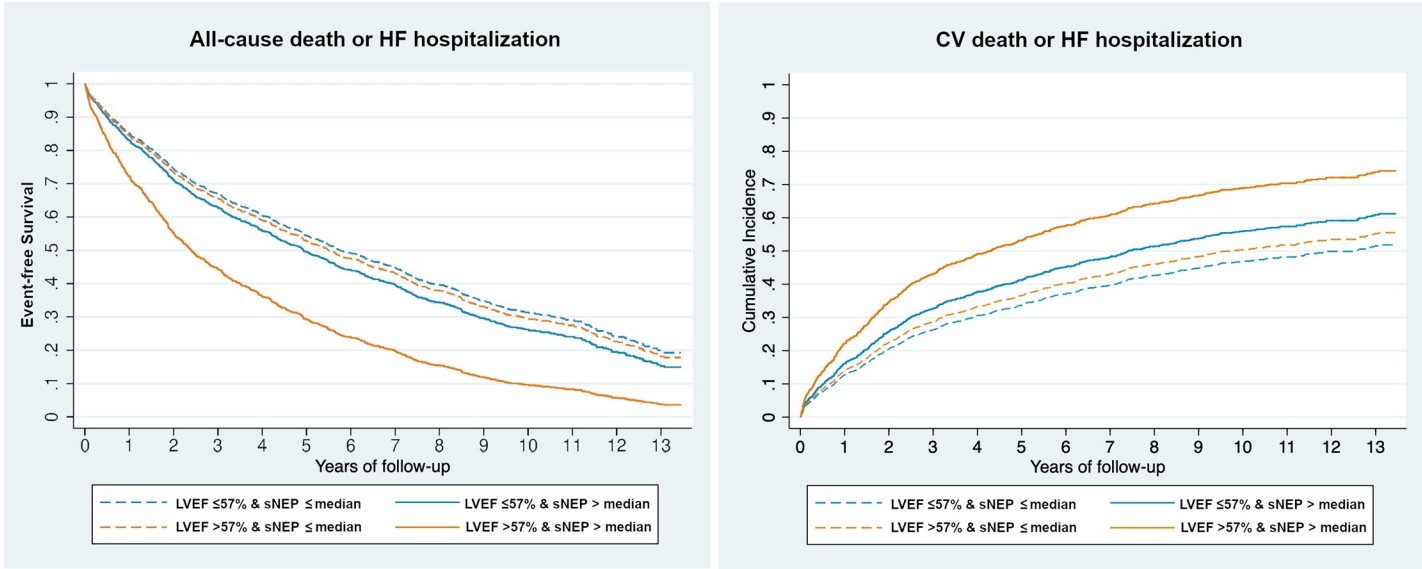

**Fig 2. Age- and sex- adjusted event-free survival and incidence curves for every LVEF group, based on sNEP levels above/below the median. Left Panel (A)** Primary composite endpoint of all-cause death or HF hospitalization; **Right panel (B)** Secondary composite end-point of cardiovascular death or HF hospitalization. Dashed lines, patients with sNEP levels below the median; solid lines, patients with sNEP levels above the median. Median values, 0.601 ng/mL in patients with LVEF >57% (blue lines) and 0.556 ng/mL for patients with LVEF ≤57% (orange lines).

(cohort selection), pre-analytical (serum vs. plasma+EDTA, freezing temperature, freeze–thaw cycles, etc), and analytical issues (the epitope explored by the different commercially available assays is uncertain) [10].

The accurate quantification of sNEP remains a challenge that needs to be overcome to suppress potential biases regarding the interpretation of the different studies [12]. sNEP blood concentrations varied exaggeratedly among studies [4, 9, 11]. sNEP concentrations were 3–10 fold higher in Lyle et al. (3.5 ng/mL in HFpEF patients and 8.5 ng/mL in controls) and Goulash et al. (2.86 ng/mL in HFpEF patients) studies than in our cohort. The assay used in our studies displays 0% cross-reactivity with the two metallopeptidases most similar to this sequence, namely endothelin-converting enzymes (ECE) 1 and 2, and also does not display cross-reactivity with erythrocyte cell-surface antigen (KELL), another protein with strong homology with NEP [17]. Furthermore we previously reported that with our quantification method circulating sNEP detected was catalytically active [13]. So we are confident that what we are measuring is actually involved in the pathophysiology of really HFpEF patients. Beyond the issue of the method, using 50% as the cut-off for HFpEF is arbitrary and eventually the range of 50–55% LVEF may represent–at least in some patients–an incipient degree of systolic dysfunction rather than a true HFpEF phenotype. Indeed "normal" LVEF by 2D echocardiography is probably nearest to 55% than 50% [18, 19] and most LVEF studies are usually performed by 2D echocardiogram.

Up to or knowledge sNEP levels have not been evaluated for diagnostic purposes in HF. With the hypothesis that higher sNEP levels would correlate with lower natriuretic peptide levels, worse diastolic function, and subsequent clinical incident HFpEF, Reddy et al. [20] performed a population study with 1,536 participants from Olmsted County, Minnesota. The authors found that low sNEP was paradoxically associated with worse diastolic dysfunction and hypertension but not with outcomes, including incident HF over a median of 10.7 years of follow-up.

As mentioned previously, the stronger prognostic value of sNEP levels in patients with HFpEF and LVEF >57% opens the door to personalized treatment in those patients, with the

possibility that the inhibition of NEP with sacubitril/valsartan might be beneficial in these patients. We realize that this hypothesis is speculative and that it should be confirmed in prospective clinical trials, but it seems interesting enough to be tested, giving the fact that HFpEF remains nowadays orphan of proven therapeutics. Eventually, assessment of sNEP in the patients included in the PARAGON-HF study could bring light into this. We learned that biomarker-guided management of patients might not be as effective as desired [21], but treating to achieve a target biomarker blood concentration might not be the same that selecting patients for receiving an specific treatment. This approach might bring closer HFpEF treatment to effective target selective treatments achieved in oncology.

## Limitations

Technical limitations of the assay should be acknowledged. Only circulating sNEP was assessed, but in humans NEP is widely expressed in several organs, and thus the results reported here might underrepresent overall NEP expression and activity. Furthermore, the experimental assay we used for sNEP determination has long incubation times and it is not ready for clinical use. Up to our knowledge there are no marketed immunoassays approved in clinical practice.

Due to the worse prognosis observed, we cannot discard a selection bias in our patients with LVEF >57%.

## Conclusions

sNEP prognostic value in patients with HFpEF and LVEF >57% outperforms that observed in patients with lower LVEF. These data support the personalized use of sNEP in HFpEF patients that may benefit from treatment with sacubitril/valsartan. It would ideal to conduct a prospective, randomized trial in HFpEF patients and ARNI treatment based sNEP measuring.

## Supporting information

**S1 File. SPSS study dataset.**
(SAV)

**S2 File. Excel study dataset.**
(XLSX)

## Acknowledgments

We wish to thank the nurses in the HF unit, Beatriz González, Carmen Rivas, Patricia Velayos, Ana Pulido, Eva Crespo and Violeta Díaz, for data collection and their invaluable work in the unit. We also thank physicians Marta de Antonio and Pedro Moliner for their priceless work in the unit. We also wish to acknowledge Redes Temáticas de Investigación Cooperativa en Salud (RETICS) and Red Cardiovascular (RD12/0042/0047).

## Author Contributions

**Conceptualization:** Josep Lupón, Antoni Bayes-Genis.

**Data curation:** Evelyn Santiago-Vacas, Germán Cediel, Pau Codina, Mar Domingo, Elena Revuelta-López, Giosafat Spitaleri, Javier Santesmases.

**Formal analysis:** Evelyn Santiago-Vacas, Germán Cediel, Elisabet Zamora, Julio Núñez, Antoni Bayes-Genis.

**Methodology:** Josep Lupón, Germán Cediel, Elena Revuelta-López.

**Writing – original draft:** Josep Lupón.

**Writing – review & editing:** Josep Lupón, Evelyn Santiago-Vacas, Germán Cediel, Pau Codina, Mar Domingo, Elena Revuelta-López, Elisabet Zamora, Giosafat Spitaleri, Javier Santesmases, Julio Núñez, Antoni Bayes-Genis.

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
