## [Decision Letter · Decision Letter 0]

24 Feb 2021

PONE-D-21-00228

Circulating neprilysin hypothesis: a new opportunity for Sacubitril/Valsartan in patients with heart failure and preserved ejection fraction?

PLOS ONE

Dear Dr. Lupón,

Thank you for submitting your manuscript to PLOS ONE. After careful consideration, we feel that it has merit but does not fully meet PLOS ONE’s publication criteria as it currently stands. Therefore, we invite you to submit a revised version of the manuscript that addresses the points raised during the review process.

Both of the reviewers regard this paper important, but the author had better to show how many patients screened to get to their numbers, and if serum was collected and stored, when was it analyzed? 

We look forward to receiving your revised manuscript.

Kind regards,

Yoshiaki Taniyama, MD, PhD

Academic Editor

PLOS ONE

Journal Requirements:

"A. Bayes-Genis has received speaker fees from Novartis; Julio Núñez has received speaker fees from Novartis, Vifor Pharma, Boehringer Ingelheim, Astra Zeneca, Rovi, and Novonordisk; A. Bayes-Genis and J. Lupón have applied for a patent for sNEP as prognostic biomarker and report a relationship with Critical Diagnostics. The rest of the authors have no conflicts of interest."

Reviewers' comments:

Reviewer's Responses to Questions

**Comments to the Author**

1. Is the manuscript technically sound, and do the data support the conclusions?

Reviewer #1: Partly

Reviewer #2: Yes

2. Has the statistical analysis been performed appropriately and rigorously? 

Reviewer #1: Yes

Reviewer #2: Yes

3. Have the authors made all data underlying the findings in their manuscript fully available?

Reviewer #1: Yes

Reviewer #2: No

4. Is the manuscript presented in an intelligible fashion and written in standard English?

Reviewer #1: Yes

Reviewer #2: Yes

5. Review Comments to the Author

Reviewer #1: 1- Hypothesis properly outlined in the Introduction

2- Methods: Were samples collected on patients, stored and then all analyzed with the same assay at once or analyzed over a 10-year period? sNEP is not an assay normally processed in regular hospital laboratories. This needs to be clarified before accepting for publication

3- CV death: defined by the authors. Appropriate follow up and tracking of events/outcomes.

4- Results: Can the authors provide the number of patients screened for this study? Unclear how many patients screened – must be clarified prior to publication

5- Table 1: demographics are consistent with previous studies (HFPEF vs. HFREF)

6- Hidralazine (Table 1) should be spelled hydralazine

7- Table 2: Data similar to other studies i.e. worse outcomes in patients with HFREF vs. HFPEF

8- Discussion: appropriate comments made on results. Discussion of limitations of testing

Reviewer #2: Lupon et. al. present results from a large cohort of patients with HF hospitalized over a 10-year period. sNEP is measured and associations are examined among those with and without preserved EF (>=57%) and above and below median of sNEP. The association between sNEP and outcomes in HFpEF patients warrants further external validation. Further, whether only those who have elevated sNEP will benefit from ARNI will require further studies. These limitations are nicely outlined.

Additional investigations could consider:

-AUC for sNEP in patients with and without pEF

-Remove either the event-free survival or cumulative incidence as this information is redundant in the figure

-Add discussion about whether sNEP could be used before HFpEF develops to predict risk for HF

6. PLOS authors have the option to publish the peer review history of their article (what does this mean?). If published, this will include your full peer review and any attached files.

Reviewer #1: No

Reviewer #2: No

---

## [Author Response · Author response to Decision Letter 0]

2 Mar 2021

Reviewer #1: 

1- Hypothesis properly outlined in the Introduction.

Thank you.

2- Methods: Were samples collected on patients, stored and then all analyzed with the same assay at once or analyzed over a 10-year period? sNEP is not an assay normally processed in regular hospital laboratories. This needs to be clarified before accepting for publication.

We appreciate your comment and apologize for the generated doubts. Blood samples were collected and stored at -80°. Afterwards they were processed in two time-periods: 1069 patients in June-July 2014 and the rest in November 2018. We clarify this aspect in the methods section: “…Blood samples were obtained between 09:00 am and 12:00 pm and stored at at -80º and analyzed without previous freeze-thaw cycles. Analyses were performed in two time periods: June-July 2014 in the first 1069 patients and November 2018 in the rest”.

3.-CV death: defined by the authors. Appropriate follow up and tracking of events/outcomes.

Thank you.

4- Results: Can the authors provide the number of patients screened for this study? Unclear how many patients screened – must be clarified prior to publication.

Thank you for the comment. We clarify this question in the revised version of the manuscript: “Circulating sNEP was measured in 1,428 patients with HF who were consecutively enrolled in the study from May 2006 to February 2016, out of the 1,765 patients who were attended during this period time. No clinical criteria for exclusion were established and only consent and availability of blood sample determined the included patients”.

5- Table 1: demographics are consistent with previous studies (HFPEF vs. HFREF).

We agree with the comment.

6- Hidralazine (Table 1) should be spelled hydralazine

Done.

7- Table 2: Data similar to other studies i.e. worse outcomes in patients with HFREF vs. HFPEF.

In the revised version of the manuscript we added the following sentence in the limitations section: “…Due to the worse prognosis observed, we cannot discard a selection bias in our patients with LVEF >57%”.

8- Discussion: appropriate comments made on results. Discussion of limitations of testing.

We appreciate your comment. In the revised version of the manuscript we added the following limitation: “…Technical limitations of the assay should be acknowledged. Only circulating soluble NEP was assessed, but in humans NEP is widely expressed in several organs, and thus the results reported here might underrepresent overall NEP expression and activity. Furthermore, the experimental assay we used for sNEP determination has long incubation times and it is not ready for clinical use. Up to our knowledge there are no marketed immunoassays approved in clinical practice.

Due to the worse prognosis observed, we cannot discard a selection bias in our patients with LVEF >57%..”

Reviewer #2: Lupon et. al. present results from a large cohort of patients with HF hospitalized over a 10-year period. sNEP is measured and associations are examined among those with and without preserved EF (>=57%) and above and below median of sNEP. The association between sNEP and outcomes in HFpEF patients warrants further external validation. Further, whether only those who have elevated sNEP will benefit from ARNI will require further studies. These limitations are nicely outlined.

Additional investigations could consider:

-AUC for sNEP in patients with and without pEF.

Thank you for suggestion. We choose Harrell’s C-statistic because it takes into account time-to-event, although it always gives lower numbers that AUC by logistic regression. In the revised version of the manuscript we added:

“Harrell’s C-statistic for sNEP above/below the median, taking also in consideration age and sex, was 0.730 (0.682-0.779) in patients with LVEF >57% and 0.677 (0.664-0.776) in patients with LVEF ≤57% for the primary composite end-point, and 0.704 (0.643-0.765) and 0.629 (0.603-0.655), respectively, for the secondary composite end-point of cardiovascular death or HF hospitalization)”.

-Remove either the event-free survival or cumulative incidence as this information is redundant in the figure.

We apologize for the lack of clarity. As mentioned in the figure legend, left panel (A) represents event-free survival for the composite end-point of all-cause death or HF hospitalization, while right panel (B) represents cumulative incidence of the composite end-point of cardiovascular death or HF hospitalization (with competing risk taken into account in the analysis). We clarify this in the reviewed figure.

-Add discussion about whether sNEP could be used before HFpEF develops to predict risk for HF

Thank you for your recommendation. In the revised version of the manuscript we added the following paragraph in the Discussion section: 

“Up to or knowledge sNEP levels have not been evaluated for diagnostic purposes in HF. With the hypothesis that higher sNEP levels would correlate with lower natriuretic peptide levels, worse diastolic function, and subsequent clinical incident HFpEF, Reddy et al. [20] performed a population study with 1,536 participants from Olmsted County, Minnesota. The authors found that low sNEP was paradoxically associated with worse diastolic dysfunction and hypertension but not with outcomes, including incident HF over a median of 10.7 years of follow-up.”

[20] Reddy YNV, Iyer SR, Scott CG, Rodeheffer RJ, Bailey K, Jenkins G, et al. Soluble Neprilysin in the General Population: Clinical Determinants and Its Relationship to Cardiovascular Disease. J Am Heart Assoc. 2019; 8: e012943. doi: 10.1161/JAHA.119.012943.

---

## [Decision Letter · Decision Letter 1]

23 Mar 2021

Circulating neprilysin hypothesis: a new opportunity for Sacubitril/Valsartan in patients with heart failure and preserved ejection fraction?

PONE-D-21-00228R1

Dear Dr. Lupón,

We’re pleased to inform you that your manuscript has been judged scientifically suitable for publication and will be formally accepted for publication once it meets all outstanding technical requirements.

Kind regards,

Yoshiaki Taniyama, MD, PhD

Academic Editor

PLOS ONE

Additional Editor Comments (optional):

The author responded the problems.

Reviewers' comments:

Reviewer's Responses to Questions

**Comments to the Author**

1. If the authors have adequately addressed your comments raised in a previous round of review and you feel that this manuscript is now acceptable for publication, you may indicate that here to bypass the “Comments to the Author” section, enter your conflict of interest statement in the “Confidential to Editor” section, and submit your "Accept" recommendation.

Reviewer #1: All comments have been addressed

2. Is the manuscript technically sound, and do the data support the conclusions?

Reviewer #1: Yes

3. Has the statistical analysis been performed appropriately and rigorously? 

Reviewer #1: Yes

4. Have the authors made all data underlying the findings in their manuscript fully available?

Reviewer #1: Yes

5. Is the manuscript presented in an intelligible fashion and written in standard English?

Reviewer #1: Yes

6. Review Comments to the Author

Reviewer #1: previous concerns are addressed. Acceptable for publication. Concerns regarding the description of the study population was addressed in the revised manuscript

7. PLOS authors have the option to publish the peer review history of their article (what does this mean?). If published, this will include your full peer review and any attached files.

Reviewer #1: No

---

## [Editor Report · Acceptance letter]

1 Apr 2021

PONE-D-21-00228R1 

Circulating neprilysin hypothesis: a new opportunity for Sacubitril/Valsartan in patients with heart failure and preserved ejection fraction? 

Dear Dr. Lupón:

I'm pleased to inform you that your manuscript has been deemed suitable for publication in PLOS ONE. Congratulations! Your manuscript is now with our production department. 

Kind regards, 

on behalf of

Dr. Yoshiaki Taniyama 

Academic Editor

PLOS ONE